# Incidence of Ophthalmological Complications in NF-1 Patients Treated with MEK Inhibitors

Lena Hummel [1,†], May Ameri [2,†], Shaikha Alqahtani [3], Zsila Sadighi [3] and Nagham Al-Zubidi [4,*]

1. Department of Ophthalmology, University of Texas Medical Branch, Galveston, TX 77555, USA; lahummel@utmb.edu
2. McGovern School of Medicine, The University of Texas Health Science Center, Houston, TX 77021, USA; may.ali.a.alameri@uth.tmc.edu
3. Department of Pediatrics, Pediatric Hematology and Oncology, The University of Texas MD Anderson Cancer Center, Children's Cancer Hospital, Houston, TX 77030, USA; salqahtani@mdanderson.org (S.A.); zsila@aol.com (Z.S.)
4. Department of Head and Neck, The University of Texas MD Anderson Cancer Center, Houston, TX 77030, USA
* Correspondence: nsal@mdanderson.org; Tel.: +1-913-792-7950
† These authors contributed equally to this work and are co-first authors.

**Abstract:** MEK inhibitors (MEKi) represent innovative and promising treatments for managing manifestations of neurofibromatosis type 1 (NF1). To mitigate potential ophthalmic side effects, such as MEKi-associated retinopathy (MEKAR), patients undergoing MEKi therapy routinely receive ophthalmology evaluations. Our study aims to assess the necessity of this regular screening within a predominantly pediatric NF1 population by examining the occurrence of ocular adverse events (OAE). A retrospective study evaluated 45 NF1 patients receiving MEKi. Inclusion criteria included baseline and follow-up examinations following the initiation of MEKi therapy. At each assessment, a comprehensive eye evaluation was performed, comprising a dilated fundus examination, ocular coherence tomography of the macula and nerve fiber layer, and Humphrey visual field testing. Twenty-six patients, with an average age of 13 years (range 2–23 years) and an average follow-up duration of 413 days were included in the analysis. Three different MEKi were used: selumetinib (77%), trametinib (23%), and mirdametinib (4%). None of the patients experienced retinopathy at any point during the study. Some patients had pre-existing optic neuropathies (27%), but no instances of nerve changes occurred after commencing MEKi therapy. Four patients (15%) exhibited symptoms of dry eye, all of which were effectively managed with topical lubrication.

**Keywords:** neurofibromatosis; MAP inhibitors; MEK-associated retinopathy; optic glioma; MEK inhibitor





## 1. Introduction

Neurofibromatosis type 1 (NF1) is a hereditary autosomal dominant disorder caused by mutation of the NF1 gene, which produces the neurofibromin protein. This protein plays a key role in embryonic development, specifically in the differentiation of neural crest cells, neural cells, melanocytes, mesenchymal cells, and bone cells [1]. Hence, this genetic alteration in NF1 predisposes afflicted individuals to diverse tumors, such as benign nerve sheath tumors or neurofibromas, malignant peripheral nerve sheath tumors (MPNST) or neurofibrosarcomas, brain tumors, spinal cord tumors, and optic gliomas, as well as additional complications including intellectual disabilities and bone deformities [1]. Neurofibromin tightly regulates levels of activated RAS proteins, which in turn upregulate the downstream effector proteins that form part of the RAS/RAF/MAPK pathway, which plays a crucial role in regulating cell proliferation and facilitating tumor formation [2]. Inhibiting the mitogen-activated extracellular signal-regulated kinase (MEK) enzyme though

the use of MEK inhibitors suppresses the downstream signaling pathways and causes a decrease in tumor proliferation [2].

MEK inhibitors have been identified as a therapeutic strategy for specific presentations of NF1, such as plexiform neurofibromas [2]. Plexiform neurofibromas originate from nerve tissue, and their size and anatomical location can cause challenges in the surgical excision of these tumors [3]. Nevertheless, it is important to acknowledge that MEK inhibitors come with inherent risks. The predominant adverse effects associated with MEK inhibitors include rash, diarrhea, peripheral edema, fatigue, and dermatitis acneiform. Additionally, these drugs exhibit distinctive cardiac and ophthalmologic side effects [4,5]. Studies have revealed various ocular complications, including a reduction in visual acuity, dry eye syndrome, visual field abnormalities, panuveitis, MEK-associated retinopathy (MEKAR), and retinal vein occlusion [6]. Patients undergoing MEK inhibitor therapy for NF1 commonly receive routine surveillance for these adverse effects. Interdisciplinary collaboration among oncologists, neurologists, ophthalmologists, and other specialists is essential to ensure comprehensive monitoring and treatment. With a prevalence of approximately 1 in 3164 and birth incidence of 1 in 2662 (as found in a recent meta-analysis by Lee et al.), NF1 is considered one of the most common autosomal dominant disorders [7]. As such, we believe our study will be valuable in bettering the medical management of this relatively large patient population.

## 2. Materials and Methods

Retrospectively, we reviewed the medical records of 45 patients with NF1 on MEK-inhibitor therapy, including selumetinib, trametinib, and mirdametinib, between January 2019 and December 2022 who presented to the ophthalmology clinic, underwent an ophthalmic baseline examination, and follow-up examination after therapy initiation at a single US tertiary cancer center. Patients with no baseline ophthalmic examination and patients without follow-up examination after MEK-inhibitor initiation were excluded. This study was approved by MD Anderson Cancer Center's institutional review board (IRB). Twenty-seven patients met the inclusion criteria. We collected data on patient demographics, including age and gender, presenting symptoms, primary cancer diagnosis, MEK-inhibitor therapy type, dose, and duration, age at start of treatment, reason for stopping treatment, number and date of follow-up examination visits, and clinical outcomes. The ophthalmology examinations included subjective vision changes, best corrected visual acuity, color plates, intraocular pressure by Tono-Pen (Reichart-Avia, Buffalo, NY, USA), presence of relative afferent pupillary defect, and extraocular movements, including presence of nystagmus or misalignments. A structural eye exam was performed using a slit lamp, and any anterior exam or posterior exam findings were documented. A Humphrey visual field (HVF) test (Carl Zeiss Meditec, Dublin, CA, USA) and optical coherence tomography imaging of the macula and retinal nerve fiber layer (Heidelberg Spectralis, Frankin, MA, USA) were obtained at each visit. Changes in any of these items from baseline examination were noted. The categorical variables studied are presented as a pie chart and table. Analysis of data points included nonparametric descriptive statistics.

## 3. Results

Of the 45 patients, 26 met the inclusion and exclusion criteria. Ten patients (38%) were female, and sixteen patients (62%) were male. The average age at the start of treatment was 13 years, with a median of 15 years and range of 2 to 23 years. Nineteen patients (73%) were under 18 years of age. Notably, 50% of patients had more than one primary cancer type, with the most common being plexiform neurofibromas ($n = 22$). Other primary cancers that were being treated were optic nerve gliomas ($n = 7$), low-grade gliomas of the brain stem ($n = 6$), cutaneous neurofibromas ($n = 6$), spinal nerve sheath tumors ($n = 3$), and pilocytic astrocytoma ($n = 2$) (Figure 1).

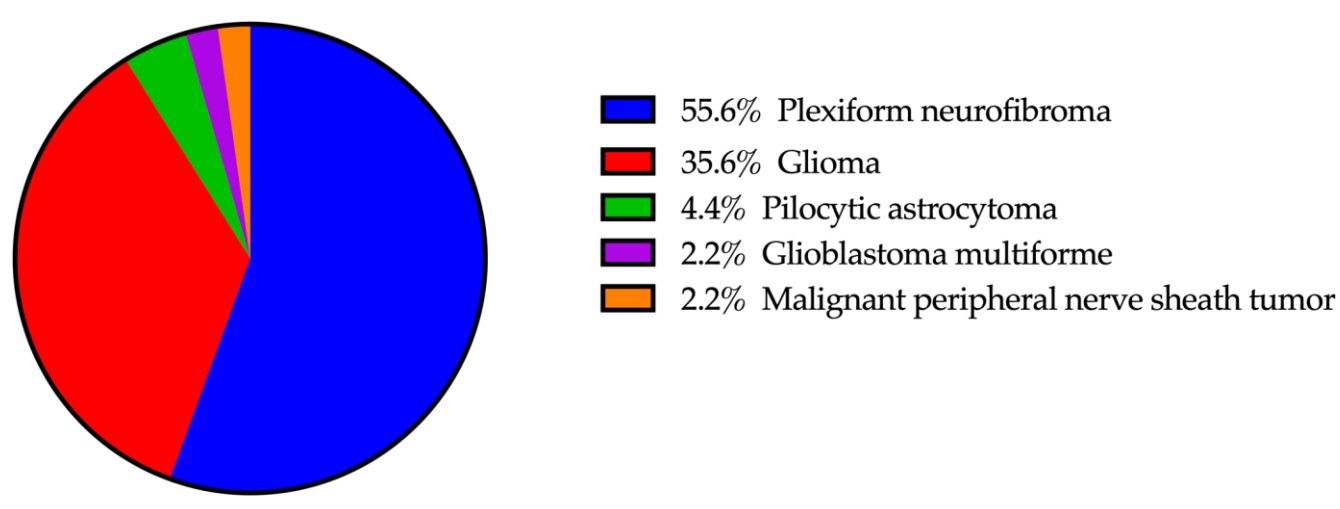

Total=45

**Figure 1.** Primary cancer type distribution.

Most of our patients were treated with selumetinib (*n* = 19, 77%), followed by trametinib (*n* = 6, 23%), and mirdametinib (*n* = 1, 4%). Two of these patients were switched to a different type of MEKi during their treatment course. One was due to adverse cutaneous side effects, and the other was due to progression of disease. Since ophthalmology only followed these patients for one of their MEK inhibitor treatments with both baseline and follow-up ophthalmology exams, the second MEK inhibitor was excluded from the analysis. These patients were followed for a mean of 413 days after starting the treatment (range 103–1122 days) for an average of 3.85 ophthalmology exams (range 2–9 total visits). No patient death was reported during the duration of the study.

At each 3–6-month ophthalmology follow-up visit, none of the twenty-seven study patients developed any retinal pathology, including retinal or sub-retinal fluid, posterior uveitis, or retinal vascular disease. Optical coherence tomography (OCT) macula showed no subretinal fluid or other retinal pathologic findings at any of the follow-up visits. On anterior exam, most patients (69%) had Lisch nodules in one or both eyes, consistent with their known NF1 diagnosis, but no other anterior segment pathology. Four patients (15%) developed signs of dry eye syndrome that had not been seen on their baseline examinations, including punctate epithelial erosions, blepharitis, fluorescein staining, and meibomian gland dysfunction. None of these patients required more than artificial tears for symptom control. Several patients had optic nerve findings, but these were documented on their baseline examinations and there were no significant changes after starting MEK inhibitors. Seven patients (27%) had a pre-existing optic neuropathy diagnosis, including optic nerve glioma, optic atrophy, and congenital glaucoma. Additionally, one patient was monocular due to history of enucleation for plexiform neurofibroma of the right eye. In none of these patients did the MEK inhibitor have to be withheld or the dose decreased for any OAEs. In one eight-year-old patient, there was consideration to hold MEK inhibitor for decreased visual acuity. OCT and HVF imaging were unable to be obtained due to a history of developmental delay and congenital nystagmus. Ultimately, the patient was continued on his medication and on follow-up exam his visual acuity had returned to baseline. A list of patients with notable pre-existing ophthalmic conditions is further detailed in Table 1.

**Table 1.** List of patients with notable pre-existing ophthalmic conditions.

| Age/Gender | Primary Diagnosis | MEKi | Starting MEKi Dose | Ophthalmological Diagnosis Prior to MEKi | Ophthalmological Exam after MEKi | Dose Changes | Tumor Response |
|---|---|---|---|---|---|---|---|
| 7 years/M | Plexiform neurofibroma (left arm) | Trametinib | 0.5 mg daily | Bilateral optic atrophy | No changes | Brief dose reduction due to dermatological adverse event | Improved |
| 6 years/M | Plexiform neurofibroma, optic tract glioma | Selumetinib | 25 mg/m² BID | Congenital nystagmus | Worse blurry vision possibly due to refractive error. * | Discontinued due to paronychia but resumed | Stable |
| 16 years/M | Plexiform neurofibroma (right orbit) | Selumetinib | 45 mg BID | Right ocular enucleation of plexiform neurofibroma | No changes | None | Stable |
| 2 years/F | Plexiform neurofibroma | Selumetinib | 25 mg/m² BID | Congenital glaucoma | No changes | None | Stable |
| 10 years/M | Optic nerve glioma, pilocytic astrocytoma | Selumetinib | 25 mg/m² BID | Bilateral optic neuropathy | No changes | None | Improved |

Abbreviation: Male: M; Female: F; MEK inhibitor: MEKi. * MEKi was held due to concern for retinopathy or disease progression. Vision changes was improved with glasses prescription.

## 4. Discussion

MEK inhibitor-associated ocular toxicities have been well documented in the literature on account of their regular implementation for multiple types of cancers. For example, MEK inhibitors are known to be a potential contributor to dry eye disease [6]. As suggested in our study, such side effects can be managed with topical therapy and do not necessitate cessation of the drug. The most common side effect, serous detachments of the neurosensory retina, also known as MEK-associated retinopathy (MEKAR), has been reported in as many as 65% of study participants in phase 1 and phase 2 clinical trials [8–10]. As such, the standard of care has been for patients to undergo baseline retinal examinations with OCT with multiple close follow-up appointments after initiating the drug. More recent studies, however, have found that these retinal findings may be relatively benign and self-limiting. Weber et al. found that while 51 patients (90%) developed subretinal fluid (SRF), only 9 (20%) were symptomatic, of which only 2 (4%) had residual SRF after treatment discontinuation. Both of these two patients had Snellen visual acuity of 20/25 or better [11]. A meta-analysis by Mendez–Martinez et al. revealed similar mild and largely asymptomatic ocular side effects of MEK inhibitors that were usually self-limiting and resolved after discontinuation or even with continued drug use. The authors suggested a reduced surveillance schedule of three initial monthly exams, then continuing follow-up visits based on the presence of SRF or discontinuation if no fluid had been appreciated [6]. Other authors suggest guidelines on the frequency of monitoring for MEK inhibitor adverse events, including ophthalmology-related adverse events, at the following interval: at the start, after 1 month, and then every 3–6 months [12]. Our negative findings support such a decrease in routine eye exams. Patients with NF1 undergoing cancer treatment are often burdened with multiple medical appointments. Dilated ophthalmology exams tend to be time consuming and are not amenable to telehealth visits, as specialized machinery is required for OCT retinal imaging. Additionally, prior studies suggest that cases of undiagnosed SRF are unlikely to harbor long-term effects, and as such, interventions such as the cessation of these potentially life-saving drugs could cause more harm overall.

Our study is unique in that our focus is exclusive to patients with NF1 undergoing treatment with MEK inhibitors. Our negative findings suggest that due to the childhood presentation of this disease, our patients are largely pediatric and may be less prone to the retinopathy experienced by the largely older oncology patients in previously published

studies. With pediatrics, there is often concern for missing vision-threatening disease in these often non-verbal patients. However, we did not find any retinal changes even in our younger patients in whom Snellen visual acuity was unable to be measured. Limitations in our study include the small study population, inconsistent follow-up duration, and the need for more long-term data to determine the length of follow-up care needed for these patients after stopping MEK inhibitors. An additional limitation includes the potential for decreased quality of Humphrey visual field tests and OCT imaging inherent when working with a largely pediatric population.

## 5. Conclusions

While eye examinations are essential in our patients who often have optic gliomas and other neurofibromatosis-related ophthalmic issues, our study results suggest that the practice of frequent surveillance retinal exams for the detection of MEK-associated retinopathy maynot be warranted, and the frequency and overall number of retinal exams may be safely decreased by extending the time between exams. As in previous studies, should SRF occur in these patients in response to their MEK inhibitor therapy, these retinal findings are likely to be mild and self-limiting despite continued MEK inhibitor treatment. Therefore, such routine examinations are unlikely to elicit a change in management but likely cause undue burden on the patient and their families. Larger cohort studies are needed to come to a final recommendation regarding the frequency of retinal exams for MEK inhibitors-associated retinopathy in NF1 patients.

**Author Contributions:** Conceptualization: N.A.-Z. and Z.S.; methodology: N.A.-Z., Z.S., L.H., M.A. and S.A.; data curation: L.H., M.A. and S.A.; writing—original draft preparation: L.H., M.A. and S.A.; writing, review and editing: N.A.-Z., Z.S., L.H., M.A. and S.A.; tables and graphs: M.A. and S.A.; supervision: N.A.-Z. and Z.S. All authors have read and agreed to the published version of the manuscript.

**Funding:** This research received no external funding.

**Institutional Review Board Statement:** The study was conducted in accordance with the Declaration of Helsinki and approved by the Institutional Review Board of MD Anderson Cancer Center (2021-0385_MOD003, 6 February 2023).

**Informed Consent Statement:** Patient consent was waived due to study exemption for secondary research on data or specimens with no consent required.

**Data Availability Statement:** The data presented in this study are available on request from the corresponding author. The data are not publicly available due to confidential patient information involvement.

**Acknowledgments:** The authors wish to thank Dianna B. Roberts, Clinical Data Mgmt. Sys. Dept. of Head and Neck Surgery, The University of Texas MD Anderson Cancer Center, for helping with statistics.

**Conflicts of Interest:** The authors declare no conflicts of interest.

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
