# Peer review of "Incidence of Ophthalmological Complications in NF-1 Patients Treated with MEK Inhibitors"

_curroncol, doi:10.3390/curroncol31050199_

Round 1

Reviewer 1 Report

Comments and Suggestions for Authors

Interesting read and surprisingly results which indeed should let us debate about current routine screening.

However is see some things that must be improved before publishing.

- in your methods you described exclusion of any preexisting ocular pathology, in your results however you describe 27% with pre existing optic neuropathy which does not go along with your exclusion critera

- what was your definition of retinopathy? Subretinal fluid? Or any kind of posterior uveitis or retinal vascular pathology? You should make make a point about any occurrence of drug induced uveitis which is not uncommon in MEK inhibitors. Was there any case or did you include these in retinopathy?

- which ophthalmology examination where part of your screening protocol? These should be included in your methods. You say nothing about fundus photography. Angiography? Tonometry?

- did you see any changes with angiography? As you stated uveitis and vascular pathology can result by these drugs and are sometimes only visible via fluorescein angiography

- you nicely spend a table about CTCAE however you do not report about these in your results. So there is no reasonable in showing that table to me.

- did you do any objective measurement of dry eye disease as it was quite common in your study group

I hope some of my critics can be answered so your manuscript can be improved which would make this interesting results more valuable.

Author Response

1. in your methods you described exclusion of any preexisting ocular pathology, in your results however you describe 27% with pre-existing optic neuropathy which does not go along with your exclusion critera

            Agree. Comment in the methods section has been deleted.

2. what was your definition of retinopathy? Subretinal fluid? Or any kind of posterior uveitis or retinal vascular pathology? You should make a point about any occurrence of drug induced uveitis which is not uncommon in MEK inhibitors. Was there any case or did you include these in retinopathy?

            There was no case of uveitis in any of our patients during follow up. The definition of retinopathy was meant to be broad and encompass any retinal pathology including the development of sub-retinal fluid or any posterior uveitis or retinal vascular pathologies. This has been edited into the manuscript.

3. which ophthalmology examination where part of your screening protocol? These should be included in your methods. You say nothing about fundus photography. Angiography? Tonometry?

            No fundus photography was obtained. In addition to the eye exam, HVF and OCT RNFL and macula were done for each visit. This has been edited into the methods section of the manuscript.

4. did you see any changes with angiography? As you stated uveitis and vascular pathology can result by these drugs and are sometimes only visible via fluorescein angiography

            No IVFA images were obtained.

5.  you nicely spend a table about CTCAE however you do not report about these in your results. So there is no reasonable in showing that table to me.

            Agree. Table and comment have been deleted.

6. did you do any objective measurement of dry eye disease as it was quite common in your study group

Dry eye was recorded based on exam documentation of “punctate epithelial erosions”, “fluorescein staining”, “meibomian gland disease”, or “blepharitis” on the anterior slit lamp exam. Further testing such as Schirmer’s testing or testing of team osmolarity for example, was not performed. Dry eye was only recorded as a possible development of MEK inhibitor therapy if it were not noted on baseline examination.

Reviewer 2 Report

Comments and Suggestions for Authors

In this manuscript, the authors present an investigation into the incidence of ophthalmological complications in NF-1 patients undergoing treatment with MEK inhibitors. The study aims to evaluate the necessity of regular screening for these complications within a predominantly pediatric NF1 population by examining the occurrence of ocular adverse events (OAE). The retrospective study included 45 NF1 patients receiving MEKi therapy, each undergoing a comprehensive eye evaluation at regular intervals. This evaluation encompassed a dilated fundus examination, ocular coherence tomography of the macula and nerve fiber layer, and Humphrey visual field testing. The analysis comprised twenty-six patients, with an average age of 13 years (range 2-23 years) and an average follow-up duration of 413 days. Notably, none of the patients experienced retinopathy during the study period. Some patients presented with preexisting optic neuropathies (27%), but no new instances of nerve changes occurred following the initiation of MEKi therapy. Additionally, four patients (15%) exhibited symptoms of dry eye, all of which were effectively managed with topical lubrication.

To enhance the manuscript, the following modifications are recommended:

1.     To provide a comprehensive background, it would be beneficial for the authors to discuss the function of the NF1 gene, shedding light on its role in the development of neurofibromatosis type 1.

2.     It would be valuable to explore the incidence of NF-1 patients in the population, providing context for the prevalence of this condition and the potential impact of MEK inhibitor treatment on a broader scale.

3.     To strengthen the study's findings, the authors could consider adding a control group of NF-1 patients who are not receiving MEK inhibitors treatment. This would enable a comparative analysis and provide further evidence regarding the specific role of MEK inhibitors in the occurrence of ocular adverse events.

Author Response

1.     To provide a comprehensive background, it would be beneficial for the authors to discuss the function of the NF1 gene, shedding light on its role in the development of neurofibromatosis type 1.

            Additional information on the neurofibromin protein, and its relationship to the development of neurofibromatosis 1 has been included in the introduction.

2.     It would be valuable to explore the incidence of NF-1 patients in the population, providing context for the prevalence of this condition and the potential impact of MEK inhibitor treatment on a broader scale.

            A meta-analysis was included regarding the incidence and prevalence of this condition and cited.  

3. To strengthen the study's findings, the authors could consider adding a control group of NF-1 patients who are not receiving MEK inhibitors treatment. This would enable a comparative analysis and provide further evidence regarding the specific role of MEK inhibitors in the occurrence of ocular adverse events.

            I agree with and appreciate this comment for future directions. We would be lacking on the additional testing, namely OCT macula testing, for NF-1 patients not on MEK inhibitors, as this is not the standard of care for these patients. Additionally, as ocular adverse events were largely non-existent (aside from mild dry eye disease), a comparison would be unlikely to yield much significance, apart from possibly showing that dry eye is equally prevalent between the two groups. 

Round 2

Reviewer 1 Report

Comments and Suggestions for Authors

Thank you for considering my critics for the revised manuscript 

Reviewer 2 Report

Comments and Suggestions for Authors

My questions have been addressed. I don't have questions.